# Molecular and Functional Characteristics of Airway Epithelium under Chronic Hypoxia

**DOI:** 10.3390/ijms24076475

**Published:** 2023-03-30

**Authors:** Sharon L. Wong, Egi Kardia, Abhishek Vijayan, Bala Umashankar, Elvis Pandzic, Ling Zhong, Adam Jaffe, Shafagh A. Waters

**Affiliations:** 1School of Biomedical Sciences, Faculty of Medicine and Health, University of New South Wales, Sydney, NSW 2052, Australia; 2Molecular and Integrative Cystic Fibrosis Research Centre (miCF_RC), University of New South Wales, Sydney, NSW 2052, Australia; 3School of Clinical Medicine, Faculty of Medicine and Health, University of New South Wales, Sydney, NSW 2052, Australia; 4School of Biotechnology and Biomolecular Sciences, University of New South Wales, Sydney, NSW 2052, Australia; 5Katharina Gaus Light Microscopy Facility, Mark Wainwright Analytical Centre, University of New South Wales, Sydney, NSW 2052, Australia; 6Bioanalytical Mass Spectrometry Facility, University of New South Wales, Sydney, NSW 2052, Australia; 7Department of Respiratory Medicine, Sydney Children’s Hospital, Sydney, NSW 2052, Australia

**Keywords:** hypoxia, airway stem cell, CFTR, cystic fibrosis

## Abstract

Localized and chronic hypoxia of airway mucosa is a common feature of progressive respiratory diseases, including cystic fibrosis (CF). However, the impact of prolonged hypoxia on airway stem cell function and differentiated epithelium is not well elucidated. Acute hypoxia alters the transcription and translation of many genes, including the CF transmembrane conductance regulator (CFTR). CFTR-targeted therapies (modulators) have not been investigated in vitro under chronic hypoxic conditions found in CF airways in vivo. Nasal epithelial cells (hNECs) derived from eight CF and three non-CF participants were expanded and differentiated at the air–liquid interface (26–30 days) at ambient and 2% oxygen tension (hypoxia). Morphology, global proteomics (LC-MS/MS) and function (barrier integrity, cilia motility and ion transport) of basal stem cells and differentiated cultures were assessed. hNECs expanded at chronic hypoxia, demonstrating epithelial cobblestone morphology and a similar proliferation rate to hNECs expanded at normoxia. Hypoxia-inducible proteins and pathways in stem cells and differentiated cultures were identified. Despite the stem cells’ plasticity and adaptation to chronic hypoxia, the differentiated epithelium was significantly thinner with reduced barrier integrity. Stem cell lineage commitment shifted to a more secretory epithelial phenotype. Motile cilia abundance, length, beat frequency and coordination were significantly negatively modulated. Chronic hypoxia reduces the activity of epithelial sodium and CFTR ion channels. CFTR modulator drug response was diminished. Our findings shed light on the molecular pathophysiology of hypoxia and its implications in CF. Targeting hypoxia can be a strategy to augment mucosal function and may provide a means to enhance the efficacy of CFTR modulators.

## 1. Introduction

The standard practice for culturing patient-derived airway epithelial cells is at ambient oxygen (O_2_) tension [1]. The ambient air at sea level (dry) consists of 20.9% O_2_, while the air in a humidified CO_2_ incubator consists of 18.5% O_2_ [2]. The O_2_ tensions within a healthy human body in vivo are much lower. The lung alveoli, which have the highest exposure to O_2_, have 13% O_2_, but the O_2_ tension drops to as low as 1% in the large intestine [2]. Many chronic respiratory diseases, such as cystic fibrosis (CF), are associated with mucosal hypoxia as O_2_ tension in the upper and lower airways may be further reduced due to mucus obstruction and airway remodeling, the latter resulting in the thickening of the airway wall and narrowing of airway lumen [3,4,5,6]. Increasing evidence has revealed hypoxia as one of the key pathogenic mechanisms of chronic rhinosinusitis (CRS), the most common upper respiratory tract disorder in CF patients. Oxygen levels were substantially reduced in the sinus cavities of patients with CRS [7]. Hypoxia has been shown to induce epithelial hyperpermeability and enhance microbial invasiveness in the nasal epithelium [8]. In addition, prominent regions of hypoxia are common and associated with inflamed and infected tissues in vivo [9,10]. These create a localized hypoxic microenvironment for pathogens, including anaerobic microorganisms [11,12,13]. *Pseudomonas aeruginosa* contributes to chronic lung destruction and is located within the hypoxic mucus plaques in the airway lumens of CF patients [13].

The first observation about the impact of culturing cells at O_2_ tensions higher than the physiological conditions was a decrease in their proliferation rate [14]. Multiple studies have validated this observation, including in primary human bronchial basal stem cells (HBECs) [15]. In vitro expansion of HBECs in 2% O_2_—to mimic the hypoxic (H) stem cell microenvironment (2–6% O_2_)— had enhanced proliferation capacity [15]. Oxygenation has also been shown to play a role in HBEC differentiation, with cultures exposed to hyperoxia (30% O_2_) demonstrating improved differentiation markers beyond those exposed to normoxia [16]. In contrast, those differentiated at hypoxia (0.5% O_2_ or submerged) demonstrated reduced ciliated cell differentiation through repression of ciliogenesis via the Notch signaling pathway [17]. To date, studies have examined airway basal stem cell expansion in hypoxia followed by normoxic differentiation (HN) or normoxic expanded basal stem cells, which were differentiated in hypoxia (NH). A chronic and continued hypoxic culture of basal stem cells and differentiation (HH) to investigate hypoxia’s molecular and functional impact on airway epithelium is lacking.

Hypoxia may require consideration for the pharmacology of CF therapies, including the CFTR modulators, as evidence of reduced CFTR expression and function with acute hypoxia (24–48 h) has been reported [18,19]. In vitro assessment of CFTR modulator efficacy in patient-derived cell models has accelerated drug regulatory processes and provided access to personalized treatment to individuals who would otherwise be ineligible for approved modulator therapy [20,21,22]. Several studies comparing in vitro and in vivo CFTR function reported a correlation between the two; however, the sample size of the studies was relatively small [20,23,24,25]. A recent study reported that patient-derived cell models could identify modulator-responsive patients but may not accurately predict the magnitude of clinical benefit [26]. The fidelity and reproducibility of patient-derived cell models in modeling the individual in vivo microenvironment remain areas that require further studies.

We studied the impact of chronic hypoxia (2% O_2_; 26–32 days) on the airway basal stem cell expansion and differentiation to pseudostratified airway epithelium. We used nasal epithelial cells (hNECs) derived from 11 pediatric participants (non-CF = 3; CF = 8) as an accessible source of mucosal epithelial cells. hNECs were expanded for 5–7 days under hypoxic and compared to matched cultures at normoxia (Figure 1). Expanded basal cells from each culture were differentiated at Air–Liquid Interface (ALI) over 21–25 days under hypoxic and normoxic conditions by implementing a crossover study design of the two oxygenation levels (Figure 1). The morphology (histology and immunofluorescence), global proteomics (LC-MS/MS) and function (barrier integrity, cilia motility and ion transport physiology–CFTR modulator drug response) were assessed.

## 2. Results

### 2.1. Chronic Hypoxia Does Not Alter the Morphology of Airway Basal Stem Cells despite a Changed Proteome

The basal stem cells expanded under low oxygen tensions (2%; hypoxic; H) and demonstrated epithelial cobblestone morphology similar to the cells expanded under normal oxygen tensions (21%; normoxic; N) (Figure 2A). The cells that expanded at hypoxia were more granular than those cultured at normoxia (arrows in insets, Figure 2A). No difference in the total cell count was observed between normoxic and hypoxic cultures from each participant, which was passaged at 90% confluence (Appendix A).

Next, we investigated cellular response to hypoxia by assessing alteration to the global proteome of basal stem cells when cultured at hypoxia compared to those at normoxia. Over 2000 proteins were identified in all samples using label-free LC-MS/MS. Robust, global proteomic responses to alteration of culture oxygen level were observed in both non-CF and CF samples. Volcano plots in Appendix A show the number of Differentially Expressed Proteins (DEPs). The commonly upregulated proteins in the cells cultured under hypoxic conditions include hypoxia-inducible protein N-myc downstream-regulated gene 1 (NDRG1), mesenchymal cell marker vimentin (VIM), an enzyme for collagen synthesis (P4HA1, P4HA2) and glucose transporter (SLC2A1) (Appendix A, Appendix A). The common downregulated proteins included glutamine synthase enzyme (GLUL), members of the S100 family proteins (S100A8, S100A9) and metabolic enzymes (AKR1B10, KYNU, EPHX1) (Appendix A, Appendix A). Pathway enrichment analysis showed similar responses to hypoxia in non-CF and CF samples. Notably, pathways related to HIF-1α signaling, adhesion and cell-matrix interaction, adaptive signaling protein translation, glucose metabolism and metabolic regulation signaling were upregulated (Figure 2B, Appendix A). Consistent with the proteomics data, we validated the presence of intracellular hypoxia, which was only detected in the basal stem cells cultured at hypoxia but not normoxia (Figure 2C). Downregulated pathways in basal stem cells cultured in hypoxia included NRF2-mediated oxidative stress response, coronavirus pathogenesis pathway, oxidative phosphorylation, and Granzyme A signaling (Figure 2B, Appendix A). These pathways modulate pro-inflammatory cytokine response [27,28,29], and their downregulation may reduce cell ability to mount an effective innate immune response. The expression of pro-inflammatory cytokine IL-8 in supernatants of cultures was investigated. IL-8 levels decreased non-significantly by ~50% in non-CF (from 20.2 ± 1.7 to 9.6 ± 2.6 ng/mL) and CF (18.4 ± 4.5 to 9.9 ± 2.8 ng/mL) cultures (Figure 2D).

### 2.2. Chronic Hypoxia Alters Global Proteome in Differentiated hNECs

Next, we assessed the impact of hypoxia on the differentiation of the normoxic and hypoxic cultured basal stem cells. Normoxic-expanded basal stem cells were differentiated in (*i*) normoxic (NN) and (*ii*) hypoxic (NH) conditions. Similarly, hypoxic-expanded basal stem cells were differentiated in (*i*) normoxic (HN) and (*ii)* hypoxic (HH) conditions (Figure 1). NN represents the standard culture with both expansion and differentiation performed at normoxia.

To further characterize the impact of prolonged hypoxia, we performed an in-depth proteomics analysis of the response to hypoxia for each condition. There were, on average, ~1900 proteins identified in all samples with quantitative LC-MS/MS, with no significant difference observed between the non-CF and CF ALI cultures. To identify significant differentially abundant proteins, a pairwise comparison of conditions was performed.

Volcano plots in Figure 3A show the number of up- and downregulated proteins for each comparison pair. The least proteome alteration was observed between cultures differentiated under the same oxygen tension; irrespective of the oxygen tension, the basal stem cells were expanded in (Comparison pair 1 and 2-Figure 3A and Appendix A). Given the small number of DEPs identified in these comparison pairs, no pathway enrichment analysis was possible (Figure 3B, Appendix A). A more pronounced proteome alteration was observed in cultures that were differentiated under different oxygen tensions, even when the basal stem cell was derived from the same culture condition and most DEPs were similar in these cultures (Comparison pair 3, 4 and 5; Figure 3A and Appendix A, Appendix A). The upregulated proteins include hypoxia-inducible protein NDRG1, secretory cell markers SCGB1A1, MUC5B, VIM, and extracellular matrix regulatory protein LOXL4, while several downregulated proteins, such as metabolic enzymes (GSTA1, ADH7, ALDH1A1, NQO1), were noted (Figure 3A and Appendix A, Appendix A). Pathway enrichment analysis showed that in cultures differentiated at hypoxia, pathways including inflammatory and immune response, adhesion and cell-matrix interaction, oxidative stress response and hypoxia signaling pathways were significantly upregulated, whereas metabolic regulation was significantly downregulated (Figure 3B, Appendix A).

### 2.3. Chronic Hypoxia Alters Structure, Morphology and Barrier Integrity in Differentiated hNECs

Our proteomics data indicated actin cytoskeleton signaling was activated in hypoxic conditions. Cell adhesion and cell-matrix interactions are fundamental to the normal structure and function of human tissue [30]. For both non-CF and CF ALI, the thickness of pseudostratified columnar epithelium of normoxic differentiated cultures (NN and HN) were similar (Figure 4A,B-non-CF: 46.5 compared to 51.6 µm and CF: 49.6 compared to 51.4 µm for HN and NN, respectively). Basal stem cells differentiated under chronic hypoxia (NH and HH) and maintained differentiation capacity. The pseudostratified epithelium appeared to be squamous or transitioning towards squamous epithelium in some regions (red rectangle, Figure 4A and Appendix A) in all non-CF and CF participants analyzed (non-CF = 2; CF = 6). The epithelium was significantly thinner (*p* < 0.05) compared to normoxic differentiated cultures (NN or HN) by approximately 20 µm, with thickness averaging 30 µm for both non-CF and CF ALI (Figure 4B).

VEGF signaling is a pathway identified to be upregulated in our hypoxic samples, which plays a key role in the hyperpermeability of airway epithelial cells under hypoxia [8]. Consistent with the morphology and thickness of ALI, barrier integrity in HN cultures was maintained. The transepithelial electrical resistance (TEER) in the HN cultures was similar to the NN (Figure 4C-non-CF: 375.1 ± 20.8 versus 445.7 ± 35.2 Ω.cm^2^; CF: 613.7 ± 40.9 versus 546.1 ± 37.3 Ω.cm^2^). Meanwhile, hypoxic differentiated cultures (NH and HH) had significantly lower TEER with a decrease of at least 40% when compared to NN and HN cultures (Figure 4C-non-CF: 240.6 ± 20.3 versus 225.9 ± 18.6 Ω.cm^2^, CF: 257.0 ± 7.8 versus 228.3 ± 5.7 Ω.cm^2^).

### 2.4. Chronic Hypoxia Induces a Secretory Phenotype in Differentiated hNECs

The ratio of the distinct cell types which differentiate from the basal stem cell and lines the conducting airways is tightly regulated [31]. Previous reports indicate hypoxic conditions inhibit the differentiation of the ciliated cells [17]. Since our proteomics data had identified an increased abundance of the secretory club (SCGB1A1) and goblet (MUC5B) cell markers, we ascertained if the reduction in ciliated cells is accompanied by a shift and increase of non-ciliated secretory cell population. The immunostaining of acetylated tubulin (Actub, ciliated cell marker) and MUC5AC (goblet cell marker) showed HN had a similar differentiation profile to NN cultures, with the majority of cells being ciliated cells and a small population of secretory goblet cells (Figure 5A and Appendix A). However, cultures differentiated at hypoxia (NH and HH) demonstrated a shift towards the secretory phenotype with a higher abundance of goblet cells and a reduced number of ciliated cells. This observation was validated by the cell type composition analysis using proteomics data which showed a trend of lower expression of ciliated cell markers in cultures differentiated under chronic hypoxia (NH and HH) and a significantly higher abundance of club/goblet cells (*p* < 0.01) (Figure 5B). In addition, a decrease in acetylated tubulin protein in hypoxic conditions was confirmed when assayed by western blot, which could account for cilia loss (Figure 5C).

### 2.5. Chronic Hypoxia Alters Cilia Structure and Function in Differentiated hNECs

Given the pronounced change in phenotype and the abundance of ciliated cells, we next investigated whether there is a hypoxic-induced alteration to the structure and function of the cilia. Hypoxic expanded basal cells, differentiated at normoxia (HN), appeared to have similar cilia length to normoxic expanded and differentiated cultures (NN) (Figure 5D, left- non-CF: 3.6 µm compared to 4.5 µm; CF: 4.4 µm compared to 4.5 µm for HN and NN respectively). Cilia length in hypoxic differentiated cultures (NH and HH) were shorter compared to NN or HN, at 2.8 µm and 2.9 µm, respectively, for non-CF (non-significant) and 3.5 µm and 3.5 µm, respectively, for CF (*p* < 0.05 for HH) (Figure 5D, left).

Cilia beating at a physiological frequency in a coordinated pattern is essential for mucociliary clearance in the airways [32]. The cilia beating frequency (CBF) of NN cultures were 6.7 ± 0.1 and 6.4 ± 0.1 Hz in non-CF and CF cultures, respectively (Figure 5D, middle). In non-CF cultures, no significant difference was observed in all culture conditions when compared against NN cultures (largest difference of 0.3 Hz) (Figure 5D, middle). Meanwhile, in CF cultures, NH and HH demonstrated a significant decrease in CBF compared to NN. CBF was slower by 0.5 Hz in NH culture and by 0.8 Hz in HH cultures (Figure 5D, middle). Cilia coordination is presented as λ^2^, whereby the higher the λ^2^, the more coordinated the cilia beating. Consistent with cilia length and frequency, HN appeared to be as coordinated as the NN, at 955.7 compared to 588.3 µm^2^ (non-CF) and 657.0 compared to 673.1 µm^2^ (CF), respectively (Figure 5D, right). The coordination of cilia beating increased significantly in NH and HH when compared to NN or HN, with an increase in λ^2^ by more than 2-fold, at 2032 and 4999 µm^2^, respectively, for non-CF (*p* < 0.01) and 1268 and 2355 µm^2^, respectively for CF (*p* < 0.01) (Figure 5D, right).

### 2.6. Chronic Hypoxia Alters Ion Channels Function in Differentiated hNECs

Previous studies reported that CFTR protein, mRNA levels and/or function were reduced during acute hypoxia in human lung epithelial cells [18,19,33]. However, these reports were limited to acute hypoxia induction (2–96 h), with a study submerging cells in media to create a hypoxic microenvironment. The cells tested were either lung cancer cells or undifferentiated normal primary bronchial epithelial cells. Therefore, they did not provide information about the CFTR regulation in CF disease under chronic hypoxic conditions. Next, we examined the hypoxia-induced changes in CFTR function; specifically, we assessed response to CFTR modulator treatment (Figure 6A). Epithelial sodium channel (ENaC) activity was significantly diminished (*p* < 0.05) in hypoxic cultures compared to normoxic cultures in both non-CF and CF ALI (Figure 6B). Amiloride-inhibited currents were reduced by at least 50% from approximately −10 µA/cm^2^ to −5 µA/cm^2^ in CF ALI. Calcium-activated chloride channel (CaCC) activity demonstrated a trend of decrease in hypoxic cultures, although statistical significance was not reached (Figure 6C). For CFTR activity, a downward trend was also observed for hypoxic cultures, particularly when normoxic expanded basal stem cells were differentiated in hypoxia (NH; Figure 6D). In non-CF cultures, normoxic differentiated cultures (NN and HN) demonstrated comparable CFTR inhibitory responses of −42.5 ± 3.4 and −48.0 ± 2.3 µA/cm^2^, respectively (Figure 6D, bottom). However, NH showed a significantly reduced CFTR response by almost 50% to −23.7 ± 4.2 µA/cm^2^ (*p* < 0.05) (Figure 6D, bottom). HH showed a non-significant trend of decrease in CFTR response to −37.3 ± 3.0 µA/cm^2^ (Figure 6D, bottom). Expectedly, no CFTR function was detected at baseline in CF cultures derived from participants with severe CFTR mutation. We thus assessed CFTR function in Trikafta-treated cultures. CFTR response in treated-CF cultures followed the same trend as the non-CF cultures, although no statistical difference was reached (Figure 6D).

## 3. Discussion

In this study, we showed that primary hNECs derived from individuals with or without CF (*n* = 11) maintain their capacity to expand and differentiate at chronic (30 days) hypoxia (2% O_2_). The morphology and viability of airway basal stem cells remained unchanged when expanded under hypoxia, which can be attributed to the upregulation of glycolysis and the unfolded protein response (UPR) pathways evident from our proteomics data. Expanding cells have a high protein folding load and require a moderately oxidative environment to sustain mature protein folding. UPR is an adaptive pro-survival mechanism that reduces protein translation and load in the presence of external stressors such as hypoxia [34,35]. In conjunction, the activation of anaerobic glucose metabolism (glycolysis) maintains the high macromolecular requirements present during cell expansion and aids in cell survival [36,37,38]. This is in line with a previous study of human bronchial epithelial cells (hBECs) successfully expanded under hypoxia (2% O_2_) [39]. The proteomics data also exhibited an increase in EIF2 signaling, which initiates reversible cell cycle arrest [15,40]. This finding is supported by previous evidence from Peters-Hall et al., which highlights the upregulation of cell cycle inhibitors, p16/p21, in hBECs expanded under hypoxia [15]. In addition, airway epithelium from CF donors, in comparison with healthy donors, was shown to contain reduced levels of proliferating-basal epithelial cells, possibly in response to noxious stimuli [41]. Interestingly, primary hBEC cultures established from these CF donors differentiated in normoxia, demonstrating that CF stem cell recovers and replicate normally outside of the CF lung microenvironment [41]. There were no differences in the degree to which UPR or glycolysis was upregulated between CF and non-CF donors, which substantiates the ability of primary hNECs to expand under chronic hypoxia (2% O_2_) irrespective of CFTR sufficiency.

We observed airway basal stem cells’ proteome plasticity as differentiation of hypoxic-expanded stem cells at normoxia-generated epithelium with similar molecular, structural, and functional characteristics to normoxic-derived stem cells. Yet, basal stem cells tolerated differentiation at chronic hypoxia but exhibited profound alterations to those differentiated at normoxia. As expected, over-represented proteins in hypoxic conditions were mostly associated with oxidative stress and hypoxia signaling pathways. In turn, pathway mapping showed a decrease in the abundance of proteins involved in translation and metabolic processes. Translation and metabolic regulation are pivotal for inducing the adaptive stress response of epithelial cells to environmental hypoxia by regulating gene expression [42,43]. Global shutdown or reprogramming of translation prompts recovery from stress and cell death. We did not observe a greater adaptation to hypoxia between non-CF and CF cells, notwithstanding the CF cells’ inherent exaggerated pro-inflammatory responses. Knowledge about the perturbed pathways and function can help to understand how epithelial cells adapt to oxygen stress. Given the important role hypoxia plays in the pathogenesis of CF, identification of hypoxia-associated prognostic markers may aid in the better clinical management of patients with CF. Hypoxia-induced overexpression of NDRG1 is recognized as a prognostic biomarker in various human cancers [44,45,46]. As NDRG1 was identified as a strong indicator of hypoxic differentiated epithelial cells in our study, it may be interesting to assess its utility as a clinical biomarker.

Our proteomics data revealed that the expansion and differentiation of primary hNECs in chronic hypoxia (2% O_2_) activated actin cytoskeletal signaling in CF donors but not non-CF donors. Actin cytoskeletal signaling is important for cell growth and integrity and shapes cell differentiation [47,48]. Hypoxia (1% O_2_) has been shown to increase actin-binding proteins and cytoskeletal rearrangement markers in lung epithelial carcinoma cells [49]. In hBECs, F508del CFTR was shown to alter cytoskeletal organization and integrity, which increases pro-inflammatory signaling [50]. Despite retaining the capacity for mucociliary differentiation, the hypoxic-differentiated cultures exhibited significantly thinner epithelial layers, diminished barrier integrity and lineage segregation to a pro-secretory epithelial phenotype. The epithelial cells appeared flatter rather than having normal cylindrical columnar shape, suggesting transitioning towards squamous metaplasia. This altered epithelial architecture has been reported in explanted bronchial tissue of CF participants and smokers with COPD [51,52]. Hypoxia was also previously shown to decrease TEER in hNECs hence increasing epithelial barrier permeability via HIF-1α and vascular endothelial growth factor (VEGF). Both proteins were found to be overexpressed in the sinus mucosa of patients with sinusitis [8]. VEGF is a pro-angiogenic protein commonly reported to be increased in a variety of inflammatory conditions associated with hypoxia, including bronchial asthma and allergic rhinitis [53,54]. An increase in mucus-secreting cells was evident in both immunofluorescence and proteomics data. This is consistent with a previous study showing significant downregulation in Forkhead box J1 (FOXJ1) expression and ciliated cell differentiation in HBECs subjected to submerged culture or differentiated at hypoxia [17]. No difference was observed when comparing non-CF and CF ALI of the same conditions. A shift from the ciliated epithelial phenotype to the secretory phenotype has been reported in ex vivo and in vitro airway epithelium in the end-stage CF, which is associated with severe mucus plugging and restricted oxygen exposure requiring lung transplant [55]. Using single-cell transcriptomic analysis, proximal airway epithelium from CF donors has also been reported to contain epithelial cells transitioning into specialized secretory subsets [41], which supports the idea that the CF lung is phenotypically more similar to hypoxic-differentiated cultures shown in this study.

Goblet cell hyperplasia is a common feature of epithelium in individuals with other chronic respiratory conditions with reduced lung function, such as COPD and asthma, as well as in ALI cultures subjected to hypoxia using the submersion method [17,56,57]. The exact mechanism of this change in phenotype is not well understood but is reported to be attributed to the Notch, HIF-1α and HIF-2α signaling [17,58,59]. Hypoxia-inducible signaling promoting goblet cell hyperplasia and mucus hypersecretion by HIF-1α increasing expression of MUC5AC in airway epithelium has been reported [60]. However, HIF protein is relatively unstable and is degraded within minutes under normal O_2_ conditions [61]. In our study, HIF-1 protein was not detectable both in proteomics and western blotting, and HIF-2 expression was variable in CF and non-CF samples. Under hypoxia (2.5% O_2_), HIF-1α is only transiently stabilized and can still be degraded by the receptor for activated C-kinase 1 (RACK1) [61]. Under hypoxia-mimetic conditions, HIF-1α protein expression is significantly reduced in bronchial epithelial cells derived from a CF patient with CFTR^ΔF508/W1282X^ compared to CFTR-corrected S9 cells [62]. The lungs were shown to have higher HIF-2α expression than other major organs under hypoxia (8% O_2_), in Sprague–Dawley rats [63]. In addition, mice airway epithelium under hypoxia (10% O_2_) was found to regulate secretory club cell proliferation through HIF-2α expression [59]. More recently, hypoxic-HIF signaling has been indicated to stimulate the differentiation of airway stem cells into pulmonary neuroendocrine cells, which possess a secretory phenotype and affect periciliary fluid composition [64,65]. In hypoxia-induced lung epithelial carcinoma cells, knockdown of HIF-2α and not HIF-1α reduced cytoskeletal rearrangement markers, suggesting the involvement of HIF-2α in hypoxia-induced cytoskeletal remodeling [53]. Despite these findings, prolonged hypoxia and chronic UPR activation over 25 days, leading to reduced protein translation may explain the lower HIF-2α expression in non-CF samples [34,35].

We also showed that hypoxic differentiation resulted in the formation of shorter cilia with reduced beating frequency and enhanced coordination. Ciliary length is known to increase over time in in vitro cultures, and longer cilia length correlates with greater total bending and has the ability to sweep across longer distances [66]. The average stroke force was also shown to increase as a function of ciliary length [66]. Furthermore, at least 25% density of active cilia was required to generate mucus transport in ALI cultures (swirling) [67]. We hypothesize that the shorter cilia in hypoxic differentiation do not engage the mucus as effectively, which allows it to accumulate and build larger mucus, as indicated by the increased coordination length. The shorter and less abundant cilia in hypoxic-differentiated cultures, albeit more coordinated, suggest the likelihood of a less effective mucociliary clearance compared to cultures differentiated at normoxia. The dysfunction may contribute to a vicious cycle of stagnant mucus that cannot be cleared, further hypoxia niches and epithelium dysfunction.

ALI cell model of nasal epithelial cells is a valuable tool in facilitating the move towards personalizing treatment for CF. The ALI cell model recapitulates the genetic profile of the patient and provides proof-of-concept evidence for its potential to predict clinical response to therapy [20]. However, the development of patient-specific, high-fidelity in vitro cell models for the correlation of in vitro CFTR responses to in vivo measures of clinical biomarkers of disease outcomes remains a challenge [25]. The standard practice in CF labs is to culture the ALI cell model at 37 °C, 21% O_2_, and 5% CO_2_, which may not accurately represent the in vivo lung microenvironment as CF individuals have reduced lung function as they progress in age. This study demonstrated that hypoxic differentiation resulted in altered ENaC and CFTR activity. Acute hypoxia was previously shown to decrease ENaC activity in vivo in mice (8% O_2_ for 24 h) and in vitro in alveolar epithelial cells (AECs) (0.5–3% O_2_ for 1–6 h) [68]. A reduction in the expression of the ENaC subunit was observed in the apical membrane of AECs, attributed to Nedd4-2-mediated ubiquitination of EnaC leading to endocytosis and proteasomal degradation [68]. Similar findings of hypoxia reducing EnaC activity and the expression of α-, β-, and γ-EnaC subunits mRNA and protein were reported in cultured rat AECs, which could be partially inhibited by silencing HIF-2α [69]. CFTR expression was also previously reported to be downregulated under hypoxic conditions in both murine and human tissues in vivo, but no mechanistic study was performed [18,70]. Others have demonstrated that the CF airway infectious or inflammatory milieu has a major impact on CFTR function and the efficacy of CFTR modulators. In agreement with our study, airway cells infected with *P. aeruginosa* have been shown to have reduced CFTR function and response to modulators [70]. The current findings may indicate why CFTR modulators may have a more modest effect in clinic for those patients colonized with *P. aeruginosa* or exposed to chronic hypoxia. One of the main limitations of the study is the different sample sizes between CF and non-CF groups. While this limitation is overcome using appropriate post-hoc analysis, caution should be exercised when comparing the effects of hypoxia between CF and non-CF groups. Despite this, the study highlights the need to consider the oxygen tension present in cell cultures and/or patients’ lung function when assessing and comparing in vivo and in vitro CFTR responses. It is possible that the severity of CF lung disease, modulated by the airway microenvironment, is a factor in determining the patient’s clinical response to CFTR modulators.

## 4. Materials and Methods

### 4.1. Study Approval and Participants Biospecimen Collection

This study was approved by the Sydney Children’s Hospital Ethics Review Board (HREC/16/SCHN/120). Written informed consent was obtained from the legal guardian of all participants. Nasal brushings were collected from eight CF and three non-CF pediatric participants (Appendix A) by brushing the inferior nasal turbinate during investigative procedures or annual surveillance bronchoscopy. All CF patients had the common delta F508 deletion mutation.

### 4.2. Primary Human Nasal Epithelial Cell (hNEC) Culture

Primary cells obtained from nasal brushings were cultured using conditional reprogramming culture (CRC) method as described previously [71]. Briefly, collected cells were seeded with gamma-irradiated NIH/3T3 feeder cells (irrNIH) in F-media supplemented with 10 µM ROCK inhibitor Y-27632 at 37 °C, 5% CO_2_. At 80–90% confluence, co-cultures were dissociated using a differential trypsin method (Lonza CC-5034) and cryopreserved. Cryopreserved, passage 1 basal nasal cells were seeded in two plates pre-seeded with 15,000 irrNIH cells/cm^2^. One plate was cultured at 37 °C, 5% CO_2_, 21% O_2_ (normoxia, N) and the other plate was cultured in parallel at 37 °C, 5% CO_2_, 2% O_2_ (hypoxia, H) in a tri-gas incubator (New Brunswick™ Galaxy^®^ 48R; O_2_ range 1–19%). Once reaching 80–90% confluence, cells were dissociated as mentioned above and were either seeded onto transwell inserts for air-liquid interface (ALI) differentiation or frozen as cell pellets for proteomics assessment. Culture supernatants were frozen for IL-8 assessment. Media change and passaging of both normoxic and hypoxic cultures for each participant were performed simultaneously. Airway basal stem cell expansion was assessed over one passage only in this study since the characteristics of airway basal stem cells expanded over serial passages, e.g., expansion capacity and population doubling rate, have already been extensively studied [15,56].

### 4.3. Image-iT Green Hypoxia Reagent Staining

Passage 1 hNECs cultured in 96-well glass bottom plates in normoxia and hypoxia were incubated with 5 µM Image-iT reagent for 30 min at 37 °C. Cultures were then refreshed with fresh media and cultured overnight at 37 °C. On the following day, hNECs were incubated with 2 µg/mL Hoechst 33342 for 30 min, and images were acquired using Leica SP8 confocal microscope on a 63×/1.4 oil immersion objective.

### 4.4. Human Airway Epithelial Air-Liquid Interface (ALI) Differentiated Culture

Expanded hNECs were seeded onto collagen I-coated 6.5 mm transwell membrane inserts (Sigma CLS3470) at a density of 250,000 cells/insert for ALI differentiation, as described previously [71]. A crossover study design of the two oxygenation levels was implemented (Figure 1), whereby hNECs expanded under normoxic conditions and were subjected to differentiation at both normoxia (NN) and hypoxia (NH). Likewise, hNECs expanded under hypoxic conditions and were subjected to differentiation at both normoxia (HN) and hypoxia (HH). NN and HN inserts were cultured at normoxia, and NH and HH inserts were cultured in parallel at hypoxia, as mentioned above. For all inserts, hNECs were cultured submerged in PneumaCult Ex Plus media (STEMCELL Technologies 05040) for 72 h. For ALI differentiation, apical media was removed, and basolateral compartment was replaced with PneumaCult ALI differentiation media (STEMCELL Technologies 05001). Media change was performed every second day for 21–25 days. ALI cultures were washed with warmed phosphate-buffered saline (PBS) to remove excess mucus at day 14 post-air-lift.

### 4.5. Cilia Beating Frequency and Coordination

Cilia beating in mature (21–25 days post-ALI) differentiated cultures was imaged and analyzed as described previously [72]. Imaging was performed using a Nikon Eclipse Ti2-E microscope equipped with an Andor Zyla 4.2 sCMOS camera on a CFI S Plan Fluor ELWD 20×/0.45 objective. The environmental chamber was set to 37 °C, 5% CO_2_ and either 21% O_2_ (for normoxic conditions) or 2% O_2_ (for hypoxic conditions). For each participant, three inserts were imaged per condition, and for each insert, six fields of view were acquired. Image series of 1000 frames with 3 ms exposure time were acquired per field of view (ROI 512 × 512). Cilia beating frequency (CBF) was analyzed using a custom-built script in Matlab (MathWorks, Natick, MA). Image series were filtered to remove the immobile component in each pixel, and the Fast Fourier Transform (FFT) algorithm was used to compute the temporal spectrum for each pixel in the image series. The average spectrum per field of view was calculated using the average of all the single-pixel spectra. The dominant frequency (highest peak) was then identified using the Matlab function ‘findpeaks’. Cilia beat coordination was analyzed using a custom multi-DDM algorithm script in Matlab adapted from Chioccioli et al. [73]. The algorithm outputs an oscillating decay signal, whereby the decay rate measures the degree of motion coordination within the analyzed region [72]. λ^2^ is the square of the coordination length scale below which motion is well coordinated and above which motion starts losing coordination. A higher λ^2^ means cilia are coordinated over a greater spatial length scale [74].

### 4.6. Electrophysiological Analysis of ALI-Differentiated Nasal Epithelial Cells

Measurement of short circuit currents (I_sc_) in mature (21–25 days post-ALI) differentiated cultures was performed under voltage-clamp conditions using VCC MC8 Ussing chambers (Physiologic Instruments, San Diego, CA) with a basal-to-apical chloride (Cl^−^) concentration gradient, as described previously [74]. The Ringer solution in the basal chamber contained (in mM) 145 NaCl, 3.3 K_2_HPO_4_, 10 D-Glucose, 1.2 MgCl_2_, and 1.2 CaCl_2_, while the Ringer solution in the apical chamber contained (in mM) 145 Na-Gluconate, 3.3 K_2_HPO_4_, 10 D-Glucose, 1.2 MgCl_2_, and 1.2 CaCl_2_. After a 30 min stabilization of the ALI cultures, transepithelial electrical resistance (TEER) measurements were recorded. Cultures were then treated with pharmacological compounds (in order): 100 µM amiloride (apical) to inhibit epithelial sodium channel (ENaC)-mediated Na^+^ flux, vehicle control 0.01% DMSO or 10 µM VX-770 (apical) to potentiate cAMP-activated currents, 10 µM forskolin (basal) to induce cAMP activation of CFTR, 30 µM CFTR_inh_-172 (apical) to inhibit CFTR-specific currents and 100 µM ATP (apical) from activating calcium-activated chloride channel (CaCC) currents. To rescue CFTR function in CF cultures, cells were pre-incubated with 3 µM VX-445 and 18 µM VX-661 for 48 h prior to Ussing chamber measurement. I_sc_ in response to forskolin alone (no modulator treatment) was considered as baseline activity (ΔI_sc−Fsk_). Cumulative changes of I_sc_ in response to forskolin and CFTR modulator were used as the measure of total CFTR-activated currents.

### 4.7. Histology

After Ussing chamber measurements, mature ALI cultures were fixed in 4% paraformaldehyde for 30 min at room temperature. The fixed membranes were excised from transwell inserts, embedded in 3% agarose, and processed for paraffin embedding. For each participant, one insert was analyzed per culture condition, and for each insert, a total of five 4 µm sections were collected. The sections were collected at 800 µm apart in width to ensure adequate sampling of the 6.5 mm membrane. Sections were deparaffinized and stained with hematoxylin and eosin (H&E). Images were acquired using Olympus BX60 microscope equipped with a DP28 camera on a 20×/0.5 objective. A minimum of 15 random fields of view were captured per membrane. Measurement of epithelial thickness and cilia length were performed using ImageJ software (National Institute of Health, Bethesda, MD, USA).

### 4.8. Immunofluorescence Staining and Imaging

Mature ALI cultures were stained, as previously described [75]. Briefly, cells were fixed in 4% paraformaldehyde for 30 min at room temperature and permeabilized with 0.5% Triton-X in PBS on ice for 30 min. Cells were then blocked with IF buffer (0.1% BSA, 0.2% Triton and 0.05% Tween 20 in PBS) containing 10% normal goat serum for 1.5 h at room temperature. ALI cultures were incubated with primary antibodies against MUC5AC and e-cadherin (Appendix A) for 48 h at 4 °C. The following day, ALI cultures were washed with IF buffer and incubated with Alexa Fluor 488 and 555 conjugated secondary antibodies for 3 h at room temperature (Appendix A). After washing with IF buffer, cells were incubated with Alexa Fluor 647-conjugated primary antibody against acetyl-α-tubulin and DAPI for 3 h at room temperature. Membranes were excised and mounted with Vectashield Plus antifade mounting media (H-1900; Vector Laboratories, Burlingame, CA, USA). To visualize immunofluorescence (IF) signal on the whole membrane, 26 × 26 tiled images were acquired in the widefield mode using Zeiss Elyra PALM microscope (Carl Zeiss, Jena, Germany, on a 20×/0.5 objective. To visualize IF signal at high resolution, images were acquired using Leica SP8 confocal microscope (Leica Microsystems, Wetzlar, Germany) on a 63×/1.4 oil objective. Images were then processed using ImageJ software.

### 4.9. Lysate Preparation for Mass Spectrometry and Western Blot

Whole cell lysate of CRC and ALI cultures were extracted, as previously described [76]. Briefly, cells were homogenized in 100 µL of RIPA buffer (Life Technologies 89900) with protease inhibitor cocktail (Sigma 11836153001) and sonicated at 4 °C (Diagenode B01060010). Protein concentrations were determined using the Pierce BCA Protein Assay Kit. 50 µg was used for Mass spectrometry and 25 µg for Western blot.

### 4.10. Mass Spectrometry

Samples were reduced (5 mM DTT, 37 °C, 30 min), alkylated (10 mM IA, RT, 30 min), and incubated with trypsin at 37 °C for 18 h, at a 1:20 ratio (*w*/*w*). Samples were desalted with two SDB-RPS disks (Empore, Sigma Cat # 66886-U) packed in the 200 µL pipette tip as described previously [77]. Eluted peptides from each clean-up were reconstituted in 10 µL 0.1% (*v*/*v*) formic acid and 0.05% (*v*/*v*) heptafluorobutyric acid in water. Proteolytic peptide samples were separated by nanoLC using an Ultimate nanoRSLC UPLC and autosampler system (Dionex, Amsterdam, The Netherlands) and analyzed on an Orbitrap Fusion Lumos Tribrid mass spectrometer (Thermo Scientific, Bremen, Germany), as described previously [78]. A 90 min gradient was used.

### 4.11. Protein Identification and Functional Enrichment Analysis

Raw peak lists were analyzed using MaxQuant (version 2.0.3) with the Andromeda algorithm [79]. Search parameters were: ±4.5 ppm tolerance for precursor ions; ±0.5 Da for peptide fragments; carbamidomethyl (C) as a fixed modification; oxidation (M) and N-terminal protein acetylation as variable modifications; and enzyme specificity as trypsin with two missed cleavages possible. Peaks were searched against the human Swiss-Prot database (August 2022 release). Label-free protein quantification was performed using the MaxLFQ algorithm with default parameters. Downstream analysis of quantified proteomics data was performed using DEP R Package. Proteins present in at least 50% of the samples in each non-CF and CF CRC sample and in at least 25% of the samples in each non-CF and CF ALI samples were used for subsequent analysis. Differential expression analysis was performed with *p*-value cut-off of 0.05 and log fold change cut-off of 0.264. Canonical pathway analysis of differentially abundant proteins was performed using QIAGEN IPA Fall Release 2022 (Release date: Sept 30, 2022). All plots were created using a ggplot R package, and heatmaps were created using ComplexHeatmap R Package. Mass spectrometry data (dataset identifier PXD038852 and PXD038893) are available at the ProteomeXchange Consortium via the PRIDE partner repository at https://www.ebi.ac.uk/pride/archive/projects/PXD038852 and https://www.ebi.ac.uk/pride/archive/projects/PXD038893. The full list of identified proteins and differentially abundant protein analysis are available upon request.

### 4.12. Western Blotting

25 ug of prepared lysate from ALI cultures were assayed for hypoxia-inducible factor-1 alpha (HIF-1α), HIF-2α and acetylated tubulin proteins. Proteins were separated using Bolt 4 to 12%, Bis-Tris, 1.0 mm, Mini Protein Gel (Thermo Fisher Scientific EA03785PK2) at 200 V for 30 min. Wet transfer method at 20 V for 1 h at Room Temperature (RT) was used to transfer proteins onto the PVDF membrane. The membrane was then blocked in 5% non-fat dry milk in tris-buffered saline containing 0.1% Tween (TBST) for 1 h at RT. Protein bands were detected by incubating the membranes at 4 °C overnight with the following primary antibodies: anti-HIF-1α antibody (1:1000; Thermo Fisher Scientific MA1-516), anti-HIF-2α antibody (1:1000; Thermo Fisher Scientific PA1-16510) and anti-acetylated tubulin antibody (1:1000; Sigma T7451). Loading controls were detected with anti-GAPDH antibody (1:1000; Cell Signaling Technology 5174S). ECL Select detection reagent (Cytiva RPN2235) was used to visualize protein bands on the ImageQuant LAS 4000 (GE Healthcare, Chicago, IL, USA). Densitometry on the protein bands was performed using ImageJ software. All data were normalized to GAPDH loading control.

### 4.13. IL8 ELISA

Measurement of IL-8 concentrations in CRC and ALI culture supernatants was performed using an ELISA (R&D Systems, Abingdon, UK) according to the manufacturer’s instructions.

### 4.14. Statistical Analysis

The statistical analysis used for proteomic analysis is stated in the respective section. All other data are presented as bar plots (all data points shown) with mean ± standard error of the mean (SEM). Kruskal–Wallis One-way analysis of variance (ANOVA) was used to determine statistical differences. Dunn’s multiple comparisons test was used for post-hoc analysis. Statistical analysis was performed using GraphPad Prism software v9.0.1. A *p*-value of less than 0.05 is considered to be statistically significant.

## Figures and Tables

**Figure 1 ijms-24-06475-f001:**
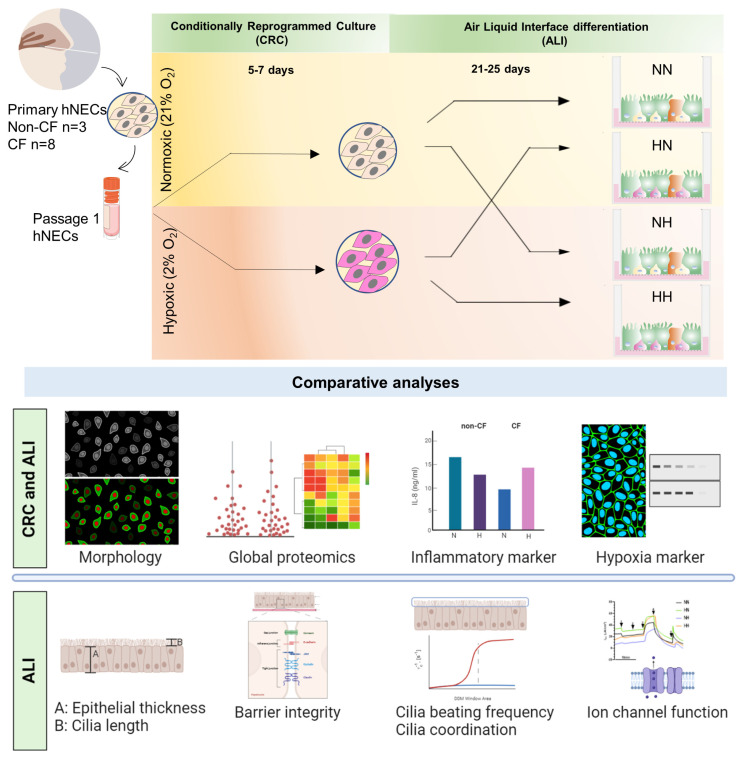
Schematic of study design. Passage 1 primary human nasal epithelial cells (hNECs) from 3 non-CF and 8 CF participants were expanded under normoxic (21% O_2_) and chronic hypoxic (2% O_2_) conditions (5–7 days). The normoxic and hypoxic derived basal stem hNECs were crossover and differentiated at air–liquid interface (ALI) at normoxia and chronic hypoxia to mature airway epithelium (21–25 days). Morphology, global proteomics, and function (inflammatory marker, barrier integrity, cilia motility and coordination, and ion transport) were compared.

**Figure 2 ijms-24-06475-f002:**
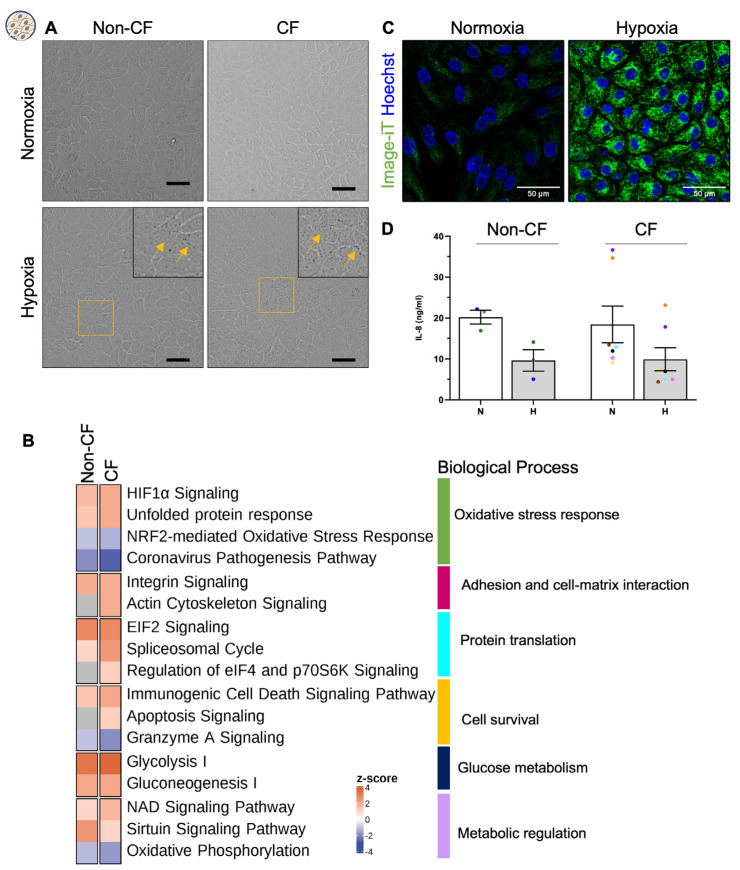
Effect of chronic hypoxia on the expansion of primary human nasal epithelial cells (hNECs). (**A**) Representative brightfield images of hNECs cultured at normoxia and hypoxia for 5–7 days. Magnified inset with yellow arrows shows granulation. Scale bars = 50 μm. (**B**) Heatmap of enriched canonical pathways of differentially expressed proteins determined by IPA. Color indicates the z-score for each pathway, with red (positive) indicating predicted activation, blue (negative) indicating predicted inhibition, and grey indicating no enrichment. Data were derived from 3 non–CF and 8 CF participants. (**C**) Representative intracellular hypoxia imaging using Image–iT green hypoxia reagent, 63×/1.4 oil immersion objective. Scale bars = 50 μm. (**D**) IL–8 ELISA measurement in culture supernatants of confluent normoxic and hypoxic hNECs. Each colored circle represents cultures of an individual participant. Data are presented as bar plots with mean ± standard error of the mean (SEM). One-way ANOVA was used to determine statistical significance.

**Figure 3 ijms-24-06475-f003:**
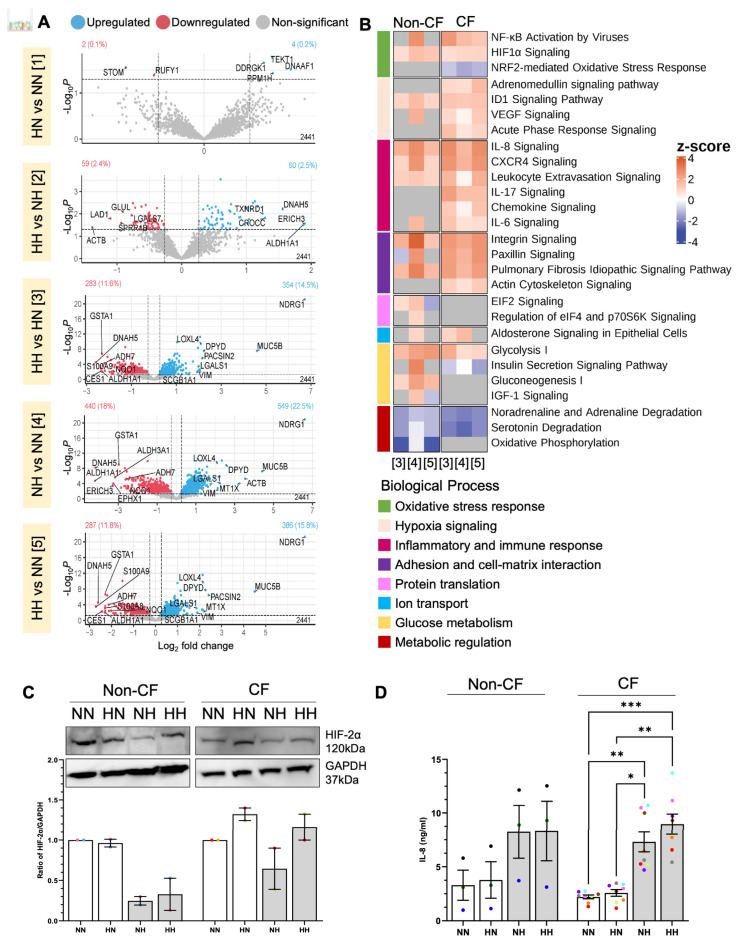
Effect of chronic hypoxia on the global proteome of differentiated human nasal epithelial cells (hNECs). (**A**) Volcano plots of differentially expressed proteins in each oxygen condition in differentiated ALI cultures from eight CF participants. Dotted lines indicate significance cut-off (*p*–value ≤ 0.05, |fold change| ≥ 1.2). The count of significantly upregulated proteins, significantly downregulated proteins and total proteins are shown in top right, top left, and bottom right, respectively. The top 5 to 10 upregulated and downregulated proteins (determined based on logFC) are labelled. Comparisons are in pairs, [1] HN compared to NN; [2] HH compared to NH; [3] HH compared to HN; [4] NH compared to NN, and [5] HH compared to NN. (**B**) Heatmap of enriched canonical pathways of differentially expressed proteins determined by IPA. Color indicates the z-score for each pathway, with red (positive) indicating predicted activation, blue (negative) indicating predicted inhibition, and grey indicating no enrichment. The columns indicate comparison 3–5 as shown in (**A**). Comparisons 1 and 2 did not result in significantly enriched pathways. Data in (**A**) and (**B**) were derived from 3 non–CF and 8 CF participants. (**C**) Western blot of hypoxia–inducible factor 2 alpha (HIF–2α) from 2 non–CF and 2 CF participants. (**D**) IL–8 ELISA of the culture supernatants of ALI cultures at days 21–25. Each colored circle represents cultures of an individual participant. Data are presented as bar plots with mean ± standard error of the mean (SEM). One-way ANOVA was used to determine statistical significance. * *p* < 0.05, ** *p* < 0.01, *** *p* < 0.001.

**Figure 4 ijms-24-06475-f004:**
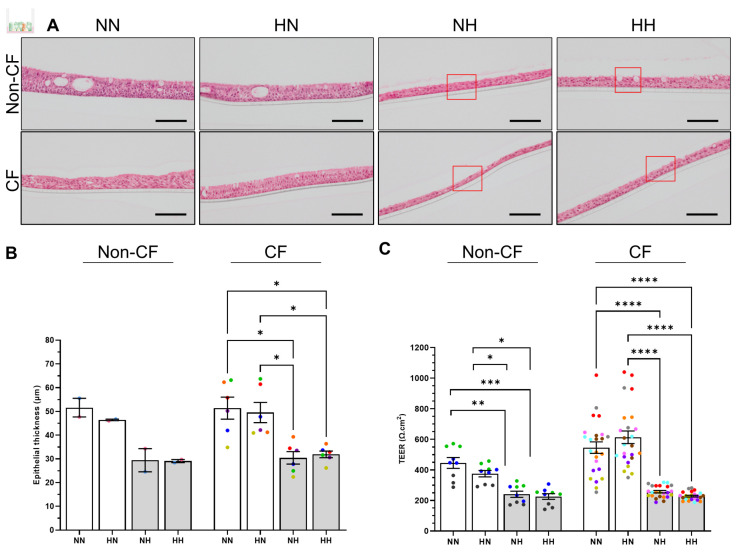
Effect of chronic hypoxia on the structure and barrier integrity of differentiated human nasal epithelial cells (hNECs). (**A**) Representative H&E stain of primary hNECS differentiated at ALI in normoxia (NN and HN) and chronic hypoxia (NH and HH) for 21–25 days. Refer to Appendix A for complete set. Red rectangle shows squamous cells or cells transitioning towards squamous morphology. 40×/0.8 objective. Scale bars = 100 μm. (**B**) ALI culture thickness measured from five sections per membrane (>20 random fields of view) per condition. Each colored circle represents an individual participant. (**C**) Transepithelial electrical resistance (TEER) of ALI cultures. Each individual participant’s data is presented with a different color. Three independent transwells were analyzed per participant. Data are presented as bar plots with mean ± standard error of the mean (SEM). One-way ANOVA was used to determine statistical significance. * *p* < 0.05, ** *p* < 0.01, *** *p* < 0.001, and **** *p* < 0.0001.

**Figure 5 ijms-24-06475-f005:**
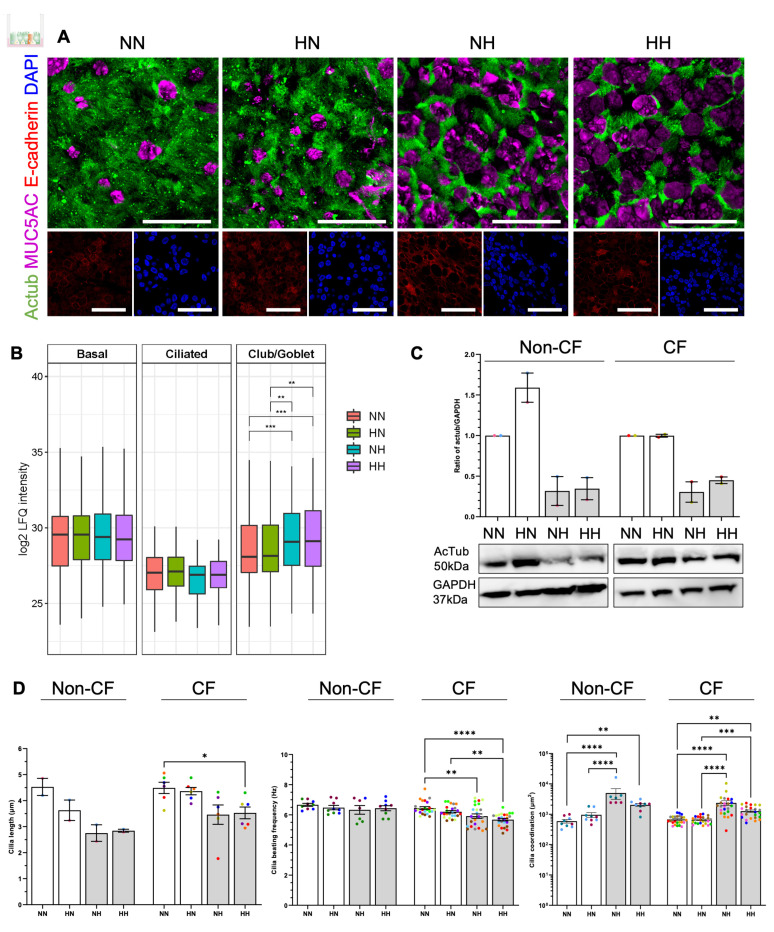
Effect of chronic hypoxia on the epithelial phenotype and cilia function in differentiated human nasal epithelial cells (hNECs). (**A**) Representative immunofluorescence staining of acetylated tubulin (green), MUC5AC (magenta) and E-cadherin (red). 63×/1.4 oil immersion objective. Scale bars = 50 μm. (**B**) Boxplots depicting the variation of log LFQ intensity in proteins found in different cell types compared across the different oxygen levels. Boxplots extend from first quartile to the third quartile, with middle line indicating the median. Upper and lower whiskers extend to the largest and smallest values within 1.5 times the interquartile range. Statistical significance of the difference in median log LFQ intensity across oxygen levels is performed using Wilcoxon Rank Sum test with Benjamini–Hochberg correction for *p*-value. * adjusted *p*-value < 0.05, ** adjusted *p*-value < 0.01, *** adjusted *p*-value < 0.001, **** adjusted *p*-value < 0.0001. Data were derived from 3 non-CF and 8 CF participants. (**C**) Western blotting of cilia marker acetylated tubulin from 2 non-CF and 2 CF participants. (**D**) Cilia length (left), cilia beating frequency (middle) and coordination (right) of mature ALI cultures differentiated at normoxia and hypoxia for 21–25 days. For cilia length, each colored circle represents an individual participant. For cilia beating frequency and coordination, each individual participant’s data is presented with a different color. Three independent transwells were analyzed per participant. Data are presented as bar plots with mean ± standard error of the mean (SEM). One-way ANOVA was used to determine statistical significance. * *p* < 0.05, ** *p* < 0.01, *** *p* < 0.001, and **** *p* < 0.0001.

**Figure 6 ijms-24-06475-f006:**
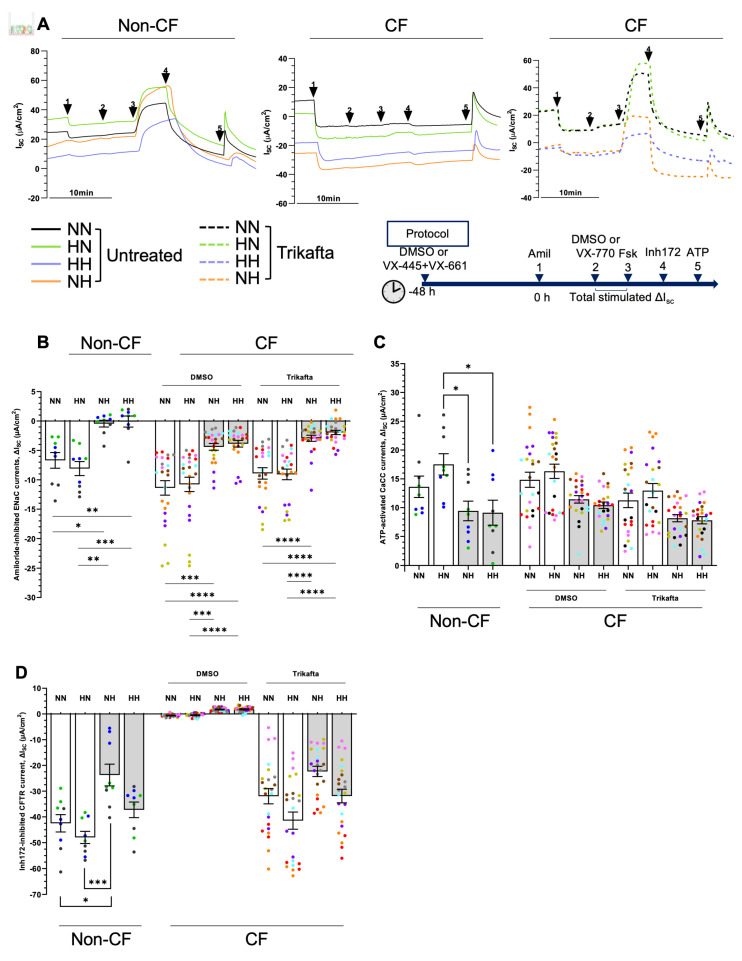
Effect of chronic hypoxia on ion transport function in differentiated human nasal epithelial cells (hNECs). (**A**) Representative Ussing chamber recordings of short circuit current (Isc) in hNECs from a non-CF and a CF participant. The protocol used to measure functional CFTR expression in hNECs in 0.01% DMSO vehicle (untreated) or pretreated with correctors (3 μM VX–445 and 18 μM VX–661 for 48 h) followed by sequential addition of 100 μM apical amiloride (1. Amil), apical addition of either vehicle control 0.01% DMSO or 10 μM VX–770 (2. DMSO or VX–770), 10 μM basal forskolin (3. Fsk), 30 μM apical CFTR inhibitor (4. CFTRinh–172), and 100 μM apical ATP (5. ATP). A basolateral-to-apical chloride gradient was used. Black line denotes NN, green line denotes HN, orange line denotes NH, and purple line denotes HH. Box plots of (**B**) amiloride-inhibited epithelial sodium channel (ENaC) currents, (**C**) ATP–activated calcium-activated chloride channel (CaCC) currents, (**D**) Inh172–inhibited CFTR current (bottom) in hNECs untreated or pretreated with VX–445 and VX–661. Each individual participant’s data is presented with a different color. Three independent transwells were analyzed per participant. Data are presented as bar plots with mean ± standard error of the mean (SEM). One-way ANOVA was used to determine statistical significance. * *p* < 0.05, ** *p* < 0.01, *** *p* < 0.001, and **** *p* < 0.0001.

## Data Availability

Data are contained within the article.

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
