# Peer review of "Molecular and Functional Characteristics of Airway Epithelium under Chronic Hypoxia"

_ijms, 2023, doi:10.3390/ijms24076475_

Round 1

Reviewer 1 Report

This review article, “Molecular and functional characteristics of airway epithelium 2 under chronic hypoxia”, reveals fascinating findings. However, the following issues and comments were raised by the reviewer and should be addressed:

1. After the initial seeding, the cells are cultivated to increase their number. In this investigation, chronic hypoxia was carried out for 26 to 30 days under hypoxic settings. This process necessitates stringent precautions to prevent oxygen contamination, while the medium culture was refreshed twice daily (line 520). After changing the media, did the author remove the cell culture from the hypoxia incubator or chamber? Is there a kit or piece of equipment used to protect cell cultures from air contamination during medium changes?

2. Please provide the brand and catalog number of the environmental (hypoxia) chamber used in this experiment (line 528).

3. Figure 1depicts a comparison analysis; what is the aim of displaying this comparative analysis? Is this the sole example of the experiment's results? Please provide further information for the morphology, global proteomics, and hypoxia marker image legends; for instance, which is CRC / ALI; what is grey or colored fluorescence?; and what are red / blue dots? What is the western blot marker?

4. Line 206-207: “The pseudostratified epithelium appeared to be compromised appearing squamous in some regions (red rectangle, Figure 207 4A) in all non-CF and CF participants analysed”. However, just one region (one red rectangle) in non-CF NH was noted by the authors. Please provide more accurate figures about this instance.

5. Figure 6A: What are the meanings of the black, green, orange, and purple lines? Please include the following information in Figure legend 6

6. Please provide citations supporting these statements (line 539-543): “The algorithm outputs an oscillating decay signal .... A higher λ2 542 means cilia are coordinated over a greater spatial length scale”.

Reviewer 2 Report

In this study the authors investigate the effects of long-term hypoxia on airway epithelium obtained from human nasal cells. A cross-over design was used to compare cells from  patients suffering from cystic fibrosis (CF) and Non-CF pediatric donors. A broad spectrum of methods was used such as proteomic analysis (by mass spectrometry), immunfluorescence combined with histomorphometic analysis and functional studies including measurement of short-circuit currents in Ussing chambers or videomicroscopical analyis of ciliary beating.
Overall the study follows a clear design and the experiments are well described. However, a great disadvantage of the study is the extreme low number of NON-CF samples (n = 3 versus n = 8 in the CF group). This weakens several of the comparisons done here between the two groups (see below).

Major comments:
- As mentioned above, group sizes are extremely different. It remains obscure why the number of Non-CF samples is so low. Usually it might be more easy to understand that you might have limitations in getting samples from patients. Considering this limitation, some sentences such as on line 253 make no sense. For example, when looking at Fig. 3D, Non-CF and CF cells show the same numeric changes, which reach statistical significance in the larger CF group but not in the small Non-CF group, i.e. the difference is a statistical artefact due to the strong difference in sample size. In some cases (such as e.g. line 195) data are only from 2 samples, which is much too low to make any conclusions.

- The statistics is not adequately described. If you compare each group against each other (e.g. Fig. 3D), you must have used a post-hoc test after the ANOVA, which must be defined in your statistical section.

- Fig. 6B: In the CF group, obviously the amiloride-sensitive current (i.e. the current carried by ENaC) is not completely downregulated by chronic hypoxia as it seems to be the case in the Non-CF group. Is this reflected by corresponding differences in the expression of ENaC (subunits) in your proteomics studies? The concentration of amiloride (100 µM) used here is extremly high and will also affect e.g. Na/H exchangers; usually a concentration of 1 - 10 µM amiloride is sufficient to suppress ENaC currents completely.

- I miss any discussion of the paradox downregulation of HIF2alpha during hypoxia (Fig. 3).

Minor comments:
- Line 132: What is the accuracy of your IL-8 assay? If you give number such as 20223 ± 1666 pg/ml, you suggest that you can determine the concentration with an accuracy of 1 pg!!! I suggest to give the numbers in ng/ml with the numbers adjusted to the accuracy of the assay (might be 20.2 ± 1.7 ng/ml). The same holds for the cilia length data. E.g. on line 245 you give a number such as 3.63 µm, suggesting that you can determine ciliar length with an accuracy of 10 nm, which is impossible with a light microscope. Adjust these number to the accuracy of your system (might give 3.6 µm in this example).

- Fig. 6D: The legend of the y-axis might be better Inh172-inhibited CFTR current.

- It should be mentioned in the main text (and not only in a Supplement table) that all patients had the (common) deltaF507 mutations.

- Line 284: define the abbreviation BH here (I assume you mean Benjamini-Hochberg correction).

Round 2

Reviewer 1 Report

The authors have addressed all reviewer comments, and the manuscript in its current form is accepted for publication in the International Journal of Molecular Sciences.